# Can the Newer Model of Breast-Specific Positron Emission Tomography Reduce the “Blind Area”?

**DOI:** 10.3390/diagnostics14182068

**Published:** 2024-09-19

**Authors:** Yoko Satoh, Jiro Ishida, Yoshitaka Inui, Akinori Takenaka, Shuji Bando, Sayuri Ishida, Hiroshi Toyama

**Affiliations:** 1Imaging Center, Fujita Medical Innovation Center Tokyo, Ota-ku 144-0041, Tokyo, Japan; 2Department of Radiology, Faculty of Medicine, Fujita Health University, Toyoake 470-1192, Aichi, Japantaki827@fujita-hu.ac.jp (A.T.); shushubnbn787@gmail.com (S.B.); kpit-nonjoke@hotmail.co.jp (S.I.); htoyama@fujita-hu.ac.jp (H.T.); 3Department of Radiology, EIZINKAI Seeds Clinic, Tokorozawa 359-1124, Saitama, Japan

**Keywords:** organ-specific PET, dedicated breast PET, blind area, photomultiplier tube-based PET, model change

## Abstract

**Objectives**: Breast-specific positron emission tomography (PET) provides higher sensitivity and spatial resolution than whole-body PET/CT, but it has a blind area. Mammary glands near the chest wall sometimes present outside the field of view (FOV). A newer, dedicated breast PET (dbPET) model has a cylindrical detector with a larger diameter than previous models, so it is expected to eliminate or reduce blind areas. This study aimed to compare breast images acquired on the new dbPET model with images acquired on an older dbPET model to evaluate blind area reduction. **Methods**: The nipple-to-chest wall distance (mm), maximum breast cross-sectional area at the FOV edge (cm^2^) on the dbPET transverse images of the scanners, and the effects of patient age and body mass index (BMI) were compared. **Results**: There was no significant difference in the nipple-to-chest wall distance between the models (*p* = 0.223). The maximum breast cross-sectional area at the FOV edge was significantly larger on the newer model’s images (*p* < 0.001). There was no significant correlation between breast size and the rate of change in both parameters. **Conclusions**: The new ring-type dbPET scanners with larger diameter detectors did not reduce the blind area observed on older dbPET scanners.

## 1. Introduction

Breast-specific positron emission tomography (breast PET) has higher sensitivity and spatial resolution than whole-body PET/CT and is a useful imaging technique for early detection and diagnosis of breast cancer [1,2,3]. Previous reports have shown that breast PET can detect breast cancer in a way that is superior to that of other imaging modalities, in certain situations [4,5]; e.g., detection of breast cancer in the dens breast on mammograms, and assessment of the effects of neoadjuvant chemotherapy compared with magnetic resonance imaging (MRI) [6,7,8,9]. However, several issues with breast PET scanners have been mentioned in previous studies. One is that the breast or chest wall is always located at the upper edge of the detector, where image noise is high, which is not the case in whole-body PET/CT. The increasing noise at the edge of the FOV is caused by the reduced sensitivity of the detector and the presence of radioactivity just outside of it. Indeed, previous studies have shown that on the clinical images, even at the edge of the FOV, the contrast between the background mammary gland and the breast lesion was high and therefore the effect of noise on lesion detection was minimal, whereas quantification was compromised in the phantom study [10,11].

Another issue that has not been resolved is “blind areas”. Depending on the size and flexibility of the breast, breast PET scans may cause a part of the mammary gland to be outside the field of view (FOV). These cases are younger, slimmer women with smaller breasts, whose mammary glands are close to the chest wall, and thus more likely to be located outside the FOV. Conversely, in older women with sufficiently large breasts, the whole mammary gland is more likely to be within the FOV, because their breasts are easily extended in the supine position. Whereas both breast MRI and breast PET are performed in the prone position, the blind area is present only with breast PET, and is the most significant limitation.

A recently developed, dedicated breast PET (dbPET) scanner has a cylindrical detector and an enlarged detector hole diameter. This design allows this new model to also be used as a dedicated brain PET by repositioning the detector [12]. Although the enlarged detector diameter has caused concern about a potential decrease in resolution in breast PET imaging, the new model is a silicon photomultiplier (SiPM)-based time-of-flight (TOF) PET scanner, which has been shown to provide the same image quality as the previous model [13]. Furthermore, the increased diameter is expected to decrease the blind area by allowing the subject’s chest wall to enter the detector hole more deeply. This study aimed to compare images acquired on both an older and the abovementioned new dbPET scanners, of women who had previously been scanned with both devices, to verify the effect of the model change on decreasing the blind area.

## 2. Materials and Methods

This retrospective study was approved by our institutional review board (HM22-542). The board waived the requirement for written informed participant consent.

### 2.1. Old and New dbPET Scanners

The old and new dbPET scanners (Elmammo™ and BresTome™, Shimadzu Corporation, Kyoto, Japan) were launched in 2014 and 2021, respectively. Notably, the new dbPET scanner has a movable detector and is capable of scanning in both breast and brain modes. A comparative study of image quality and scanner characteristics of the old and new dbPET models has been published previously [13]. The old model has a PMT-based PET detector, and a transverse effective FOV of 156.5 mm, whereas, the new model has a SiPM-based PET detector and a FOV of 162 mm. Other common characteristics of both the old and new dbPET scanners are (1) a ring-shaped detector is embedded in the surface of the bed, and the patient is placed in a prone position for scanning, with the breast hanging down into the detector hole, and (2) image reconstruction is performed by use of an attenuation correction map, obtained by extracting the breast skin from the emission data. Image reconstruction and post-filtering have been previously reported [13].

### 2.2. Patients

A total of 76 patients who underwent breast imaging with both the old and new dbPET scanners, at least once each at the EIZINKAI Seeds Clinic between December 2017 and February 2024, were included in this study. The clinical indications for dbPET included pre- (*n* = 22) or post- (*n* = 3) therapeutic evaluation of breast cancer or breast cancer screening (*n* = 51). Of the 76 patients, two with bilateral breast cancer were excluded. The ipsilateral breast of the patients (*n* = 15) who were imaged with the older dbPET scanner before treatment and with the newer one after treatment were excluded from the study. The contralateral breast of two patients after total mastectomy and the bilateral breast of one patient after partial mastectomy were included in this study. The age and body mass index (BMI) of the patient at the time of each scan were used as clinical information for the analysis. A total of 131 breasts of 74 patients were analyzed in this study.

### 2.3. Imaging Protocol

All patients fasted for ≥6 h before administration of ^18^F-fluorodeoxyglucose (^18^F-FDG) (3 MBq/kg), and underwent whole-body PET/CT scanning before dbPET scanning, 60 min after the injection. After the PET/CT scan and approximately 90 min after the ^18^F-FDG injection, dbPET scanning of each breast was performed for 7 min. During data acquisition, each scanner saved both prompt and delayed events in list-mode format, and all images were reconstructed using a three-dimensional (3D) list-mode dynamic row action maximum likelihood algorithm (LM-DRAMA), with one iteration and 128 subsets with image –space point–spread function modeling, which is also used in SiPM-based TOF-PET. In LM-DRAMA, the relaxation parameter λ depends on the subset number, and the noise propagation from the subset to the reconstructed image is suppressed as the subset number increases, resulting in fast convergence with a reasonable signal-to-noise ratio [14]. The relaxation parameter λ within the DRAMA successive approximation formula determines the amount of data reduction within one iteration, which is dependent and defined by the factor β. PET images reconstructed by LM-DRAMA are often smoothed by post-filters to suit clinical use. Based on previous findings, the old dbPET images were reconstructed with β-values of 20, a Gaussian filter with 1.17-mm full width at half maximum, and a voxel size of 0.78 × 0.78 × 0.78 mm^3^ for the post-filtering. The newer dbPET images used β-values of 100, a non-local mean with a smoothing intensity of 1.0 as a post-filter, and a voxel size of 1.1 × 1.1 × 1.1 mm^3^.

### 2.4. Image Analysis

All measurements were performed on the dbPET images displayed with a DICOM viewer. First, the shortest distance between the nipple surface and image edge of the chest wall side, defined as the “nipple-to-chest wall distance (mm),” was measured on the transverse image. Second, the maximum cross-sectional area of the breast at the edge of the FOV was simulated by the area of the ellipse calculated from the measured maximum long and short diameters of the breast on the coronal section image (Figure 1), and defined as the maximum cross-sectional area of the breast (cm^2^). It was calculated according to the following equation:$$\mathrm{Maximum}\;\mathrm{area}\;\mathrm{of}\;\mathrm{the}\;\mathrm{breast}\;\mathrm{in}\;\mathrm{the}\;\mathrm{FOV}\;({\mathrm{cm}}^{2})=\frac{b\times b^{\prime }\times \pi }{4}$$
where *b* is the long diameter and *b*′ is the short diameter (see Figure 1).

To avoid the effect of noise at the edges of the FOV on the measurement, a slice 2 mm from the FOV edge on the chest wall side was used for the measurement. PET parameters were measured twice in random order and ≥7 days apart by a nuclear medicine specialist with 18 years of experience specializing in breast imaging.

### 2.5. Statistical Analysis

The Wilcoxon signed-rank test was performed to compare the clinical characteristics of the patient between the images from the old and new scanner. Paired t-tests were used to compare the two quantitative PET parameters obtained from the old and new dbPET images. Intraclass correlation coefficients were used to evaluate the inter-reader reproducibility of the PET parameters. Values of *p* < 0.05 were accepted as indicative of statistical significance. JMP^®^ Pro 17 (17.2.0) software (SAS Institute Japan, Tokyo, Japan) was used for the analyses.

## 3. Results

Seventy-four patients were scanned with both the old and new dbPET. The mean patient age at the scan with the older scanner was 56.5 (28–82) years, with an interval of 25.5 (range: 3–62) weeks between the scans. There was no significant difference between the mean body mass index at the earlier scan (22.3 ± 3.24) compared to the later scan (22.3 ± 3.35) (*p* = 0.716).

There was no significant difference in the nipple-to-chest wall distance between the images from the old and new dbPET scanners (71.1 ± 18.3 mm and 72 ± 20.1 mm; *p* = 0.223), respectively (Figure 2A). On the other hand, the maximum breast cross-sectional area was significantly larger on the dbPET images from the new model (21.2 ± 7.2 cm^2^) than on those from the earlier model (17.9 ± 3.4 cm^2^) (*p* < 0.001) (Figure 2B).

To further confirm the effect of breast size, small and large breast size groups were selected from the patients by dividing the median nipple-to-chest wall distance measured on the old dbPET images. The results showed no significant correlation between breast size and the difference in both parameters between the old and new dbPET scanner images (Figure 3).

To compare the amount of breast included in the scan range, old and new dbPET images and a contrast-enhanced MRI image of a representative case are shown in Figure 4. In this case, the newer dbPET images had less breast volume scanned.

## 4. Discussion

This study has significant value because it is the first to compare and assess the blind areas of old and new dbPET scanners. Previous studies have all referred to the problem of blind areas in dbPET scanners, with the expectation that newer models with larger diameter detectors would improve it (Figure 5A). However, our study results showed that the expected reduction of the blind areas of the dbPET scanners has not been achieved (Figure 5B). On breast PET images, the contrast of lesion-background mammary gland FDG accumulation is high, even in dense breasts [5]. Because of this advantage, its usefulness as an additional diagnostic method to mammography screening, especially in hyperintense breasts, was described for the first time in the 2018 edition of the Japanese breast cancer practice guidelines [15]. Thus, it is younger women, often with dense breasts on mammograms, who should benefit from breast PET. However, the dilemma is that younger women are more susceptible to the negative effects of the “blind area” with breast PET.

The proposed lexicon for dbPET imaging also discusses the limitations of a blind area in dbPET systems, particularly its effect on accurate lesion detection and overall diagnostic performance [16]. Furthermore, the breast-specific PET clinical practice guidelines in Japan also caution against overlooking breast lesions present in the blind area [17]. Even here, it is suggested that technological advances in newer dbPET models will probably improve these disadvantages. In 2017, minor changes were made to the older dbPET model to bring the device closer to the breast by placing a thinner mat on the bed surface where the cylindrical detector is embedded, and by lowering the height of the plate on which the head of the woman in the prone position rests. Before and after these minor changes, it was observed that although the blind areas were reduced, the reduction was insufficient. The expectation of a reduction in the blind area with the new dbPET scanner was based on the prediction that a detector with a larger diameter would be able to scan more of the chest wall, including the breast, because of the rounded shape of the human anterior chest wall. However, the chest wall is actually not as rounded as expected in the supine position, which may explain why the blind area reduction with the new dbPET scanner could not be achieved. In contrast, cases are often observed in which the contralateral breast or parts of the upper-abdominal wall are also included in the FOV of the new dbPET, which was not been mentioned in the previous studies using the old model [2,5,8,9,10,11]. Since the dbPET scanner used in this study creates a μ-map for attenuation correction based on emission data, the presence of structures other than the breast in the FOV is inappropriate. Furthermore, the overlapping accumulation of structures other than the ipsilateral breast on the medio-lateral maximum intensity projection image may lead to misdiagnosis.

A different form of breast PET from dbPET is known as positron emission mammography (PEM), which is called the “opposite-type” because it has two parallel-facing plate detectors. PEM also has a blind area problem, but the PEMGRAPH scanner (Furukawa Co., Ltd., Tsukuba, Japan) is able to maintain image quality even when the width of the two detectors is increased [18,19]. In its scan of the medio-lateral view, the whole mammary gland is included in the FOV. The compatibility of dbPET and PEM images should be discussed in the future.

There are multiple modalities for breast cancer imaging, none of which are perfect for single use alone. The optimal modality is contrast-enhanced MRI; however, scans are difficult for patients who are claustrophobic or have metal in their bodies. Of course, it is contraindicated in patients who cannot use contrast media (patients with allergies to contrast media, asthma, or impaired renal function). Although PET tends to be requested often because of its exposure and high cost of examination, it also has limitations as an imaging modality. In particular, dbPET results in unavoidable blind areas in exchange for its high sensitivity and high spatial resolution. Until further breakthroughs occur, it is reasonable to combine dbPET with other modalities, such as ultrasonography, to account for the blind areas in dbPET scans, and the fact that some areas of the mammary glands near the chest wall may be located outside the FOV.

The ability of dbPET to detect small lesions in breasts has been shown in previous studies. Even those high-resolution PET scanners cannot detect mammary lesions located outside the FOV. If the first dbPET scan included all of the mammary gland, it would not fail to do so on the next scan. On the other hand, if a part of the mammary gland was located in the blind area on the first scan, the same defect could be expected on the next scan. In such cases, combination with other modalities should be considered from the first scan. It should also be noted that some breast cancers have low FDG accumulation even if located within the FOV. Therefore, the diagnostic radiologist should interpret the breast images comprehensively, referring not only to dbPET, but also to breast ultrasound and other imaging modalities. When breast PET is used for breast cancer screening in a health checkup, this limitation should also be disclosed to the examinee. Even though breast PET is a very painless and less stressful device for women compared to mammography, the choice should be based on a balance of advantages and disadvantages.

The usefulness of PET/CT scans for the breast in the prone position has also been reported [20]. We have experienced in practice that an additional scan in the prone position for the breast region proved to be very useful. This is one of the solutions to the problem of blind areas in breast PET. However, additional PET/CT in the prone position increases exposure due to transmission CT. This makes it difficult to use, especially in a health check-up context. Therefore, it is practical to combine breast PET with breast ultrasound, which has no exposure. One of the PEM scanners can scan the whole mammary gland with a wider distance between the two plate-like detectors, with the disadvantage of reduced spatial resolution due to the greater distance between the mammary gland and the detector [18]. The pros and cons of the two differently shaped breast PET devices should be considered and selected according to the purpose.

Our study had several limitations. First, the number of patients who were scanned with both the old and new dbPET was small. Second, because this was a retrospective observational study, there was a time lag between scans on the old and new dbPET systems. Although the patients were the same, there may have been slight differences in body shape. Third, contrast-enhanced MRI was not performed on all eligible patients, so the exact percentage of the mammary glands located in the blind area are unknown. This limitation was because the study included not only patients with breast cancer, but also women without breast cancer who had undergone breast cancer screening during health checkups.

## 5. Conclusions

Contrary to expectations, our results showed that the new ring-type dbPET scanners with larger diameter detectors did not reduce the blind area observed on older dbPET scanners.

## Figures and Tables

**Figure 1 diagnostics-14-02068-f001:**
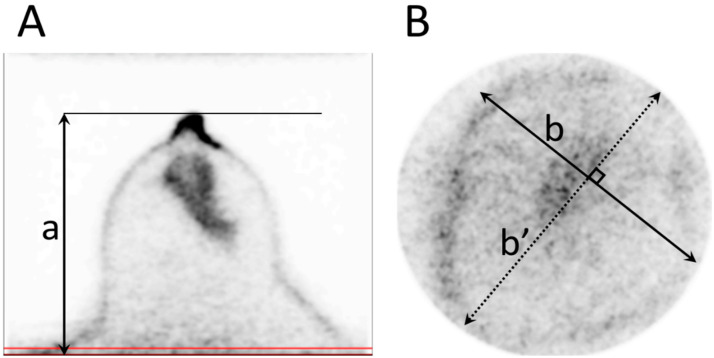
Measurements of PET parameters. (**A**) Short diameter from the nipple to the edge of the field of view (FOV) on the chest wall side (a). (**B**) The maximum area of the breast in the FOV is calculated from the long (b) and short (b’) diameters, regarding the cross-section of the breast as an ellipse. Red line indicates the slice position of (**B**).

**Figure 2 diagnostics-14-02068-f002:**
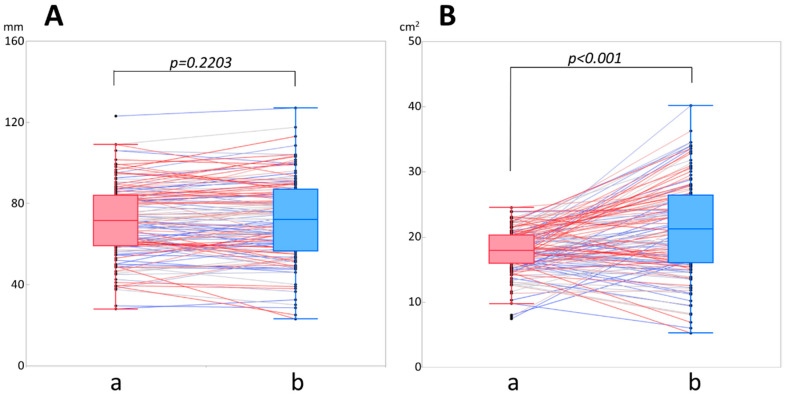
Comparison of parameters between the old (a) and new (b) dbPET scanners. (**A**) Nipple-to-chest wall distance (mm). (**B**) Maximum cross-sectional area of the breast (cm^2^). The data for the same patient in the box-and-whisker diagrams for the prior and new models are connected by a straight line. *p* < 0.005 is significant. One black bullet corresponds to one patient, and black bullets connected with a straight line represent the same patient.

**Figure 3 diagnostics-14-02068-f003:**
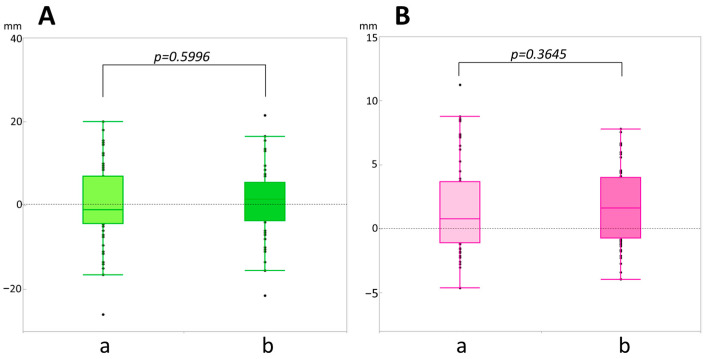
Comparison of the increase/decrease values from a new dbPET image between a smaller (a) and larger (b) breast size. (**A**) Nipple-to-chest wall distance (mm). (**B**) Maximum cross-sectional area of the breast (cm^2^). The two groups were divided by the median nipple-to-chest wall distance (mm) measured on the old dbPET images. *p* < 0.005 is significant. One black bullet corresponds to one patient, and black bullets connected with a straight line represent the same patient.

**Figure 4 diagnostics-14-02068-f004:**
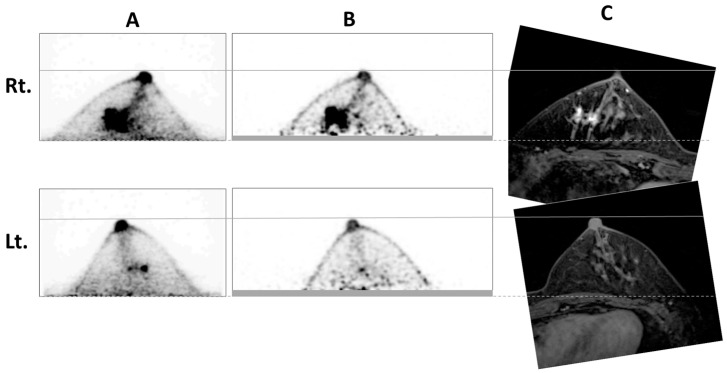
A patient in her 50s with right breast cancer. The old (**A**) and new (**B**) dbPET images and the contrast-enhanced breast MRI image (**C**). The interval between breast PETs was 3 months, and the MRIs were scanned during these months. The aspect ratio of the images was adjusted by the image viewer so that the nipples and areolas overlapped. Therefore, the breasts are the same size on these images. Rt.; right breast, Lt.; left breast’

**Figure 5 diagnostics-14-02068-f005:**
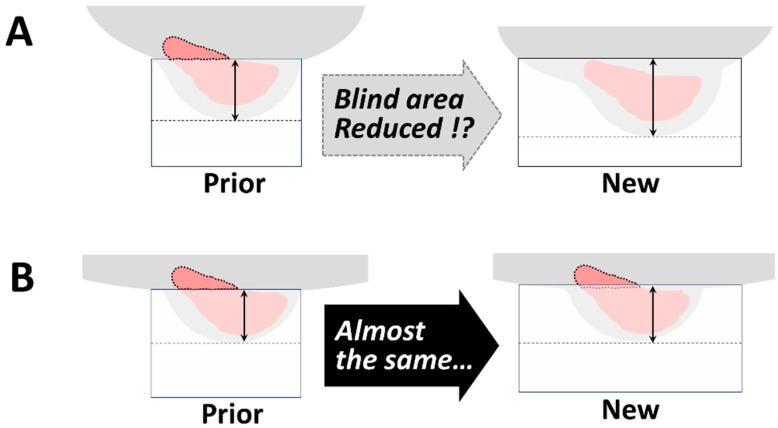
Predictions (**A**) and outcomes (**B**) of the effect of the dbPET scanner in reducing blind areas.

## Data Availability

Data are contained within the article.

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
