# Peer review of "Can the Newer Model of Breast-Specific Positron Emission Tomography Reduce the “Blind Area”?"

_diagnostics, 2024, doi:10.3390/diagnostics14182068_

Round 1

Reviewer 1 Report

Comments and Suggestions for Authors

The manuscript titled, "Can the newer model of breast-specific positron emission tomography reduce the “Blind Area”?" by Yoko Satoh et al. made a commendable effort to compare results between old and new dbPET scanners to see whether the increase in diameter of the acquisition gantry may detect the blind area.

I see this manuscript as an application of their previously published paper, Satoh Y, Motosugi U, Imai M, Onishi H. Comparison of dedicated breast positron emission tomography and whole-body positron emission tomography/computed tomography images: a common phantom study. Ann Nucl Med. 2020;34(2):119-127. doi:10.1007/s12149-019-01422-0

Although the authors have admitted to their shortcomings in the current retrospective work, The study design, acquisition, and analysis are well-written. 

The results indicated that the newer ring-type dbPET scanners, equipped with larger diameter detectors, did not spot the blind areas that could be missed in the older dbPET scanners. Yet, it is the first to compare and assess the blind areas of old and new dbPET scanners. 

In the discussion, if the authors could provide insight into what could be improved with positron emission mammography in the future to spot blind areas, this would add more value to the current manuscript.

Author Response

We sincerely appreciate your high evaluation of our research. We agree with the comment that suggestions to reduce the blind area of the breast PET should be made. We have added lines 244-245 of the main text the following text and relevant references.

The usefulness of PET/CT scans for the breast in the prone position has also been reported [20]. We have experienced in practice that an additional scan in the prone position for the breast region proved to be very useful. This is one of the solutions to the problem of blind areas in breast PET. However, additional PET/CT in the prone position increases the exposure due to transmission CT. This makes it difficult to use, especially in the health check-up situation. Therefore, it is practical to combine breast PET with breast ultrasound, which has no exposure. One of the PEM scanners can scan the whole mammary gland with a wider distance between the two plate-like detectors, with the disadvantage of reduced spatial resolution due to the greater distance between the mammary gland and the detector [18 (listed in the first edition)]. The pros and cons of the two differently shaped breast PET devices should be considered and selected according to the purpose.

  1. Nassar, L.; Kassas M.; Abi-Ghanem, AS.; El-Jebai, M.; Al-Zakleet, S.; Baassiri, AS.; Naccoul, RA.; Barakat, A.; Tfayli, A.; Assi, H.; Berjawi, G.; Estrada-Lobato, E.; Giammarile, F.; Vinjamuri, S.; Haidar, M. Prone versus supine FDG PET/CT in the staging of breast cancer. Diagnostics (Basel). 2023,13, 367. doi: 10.3390/diagnostics13030367

Reviewer 2 Report

Comments and Suggestions for Authors

Dear editor in chief,

Thank you for giving me the opportunity to review for your journal.

The manuscript under review is an interesting presentation of a study on the possibility of omitting blind area of the breast using the new model of breast dedicated PET scanners.

The study is well performed and well presented and had an important message that the blind spot may remain in the new scanners.

In my opinion the study is worth publication without any change.

Best wishes

Author Response

Thank you so much for your positive review of our paper. We are honored to have received your favorable comments.